# Are we biased on bias?
# Characterizing social bias research in the ACL community

## Abstract

Recent events in business, politics and society have shed light on the importance and potential dangers of Natural Language processing (NLP) in the real world. NLP applications have gained unprecedented popularity not just among scientist and practitioners, but also the general public. As we develop new methodologies and curate new benchmarks and datasets it is more important than ever to consider the implications and societal impact of our work. In this paper, we characterize the landscape of societal bias research within the ACL community and provide a quantitative and qualitative survey by analyzing an categorized corpus of *348* papers. More specifically, we present a definition of social bias based on ethical principals and investigate (i) types of bias, (ii) languages, and (iii) type of paper. We find that there is significantly more work on gender biases and English than other languages. Finally, we discuss the possible causes behind our findings and provide pointers to future opportunities.

## 1 Introduction

Traditionally, the NLP community has focused on ethical debates around privacy (Hovy and Spruit, 2016) ensuring that data is anonymized appropriately. More recently, there has been increased awareness that NLP research has a direct impact on peoples' lives (Mayfield et al., 2019; Bender and Friedman, 2018). For example, summarization systems can amplify misinformation (Smiley et al., 2017), and sentiment analysis (SA) systems can assign more negative sentiments/emotions based on race and/or gender (Kiritchenko and Mohammad, 2018). While such research used to be more academic (Leidner and Plachouras, 2017), these concerns are having an increasing impact in industry (Schnoebelen, 2017; Jin et al., 2021b) with consequences for users (Prabhumoye et al., 2021). It is well known that language data encodes demographics and biases (Bender and Friedman, 2018). There

is a risk that using such data can disclose inappropriate information about particular individuals, as well as undesirable attitudes towards individuals and groups (Hovy and Spruit, 2016; Eckert and Rickford, 2001) and social hierarchies (Blodgett et al., 2020). There are also concerns that systems based on inappropriate data are likely to repeat such biases, and may even amplify them (Bender and Friedman, 2018). In this paper, we survey **348** papers collected from the ACL anthology that focus on social bias and ethics in NLP research. We make three kinds of contributions, where (i) we present a working definition of social bias from a philosophy perspective, (ii) quantify our findings by annotating our corpus of papers and (iii) provide a discussion and pointers for possible future research directions. Through a quantitative analysis of current trends we attempt to answer the following questions:

- What kind of social biases is the ACL community concerned with?
- How many languages is bias studied in?
- What types of papers are present?

## 2 Related previous surveys

There is a considerable literature on social biases in NLP. Much of this work provides guidelines and/or recommendations. One of the first position papers on the topic outlined the need for ethical considerations that go beyond privacy concerns for users, and focus on the social impact of experiments and applications on individuals and (minority) groups Hovy and Spruit (2016).

Surveys on social bias emphasize a variety of different aspects, such as embedding representations, data collection and annotation, downstream task performance, metrics and limiting negative impact (Garrido-Muñoz et al., 2021; Mohammad, 2022b; Schnoebelen, 2017). Work by Bender and Friedman (2018); Hovy and Prabhumoye (2021) outline the concept of data statements and sources

of bias respectively to aid the research design process. Other research has reviewed how to mitigate bias (Chandrabose et al., 2021; Meade et al., 2022; Balkir et al., 2022), how to teach bias, ethics and privacy to students (Bender et al., 2020; Friedrich and Zesch, 2021), evaluate existing metrics (Czarnowska et al., 2021; Delobelle et al., 2022), handle challenges presented by new laws (e.g., GDPR) (Lewis et al., 2017) and apply existing principles from ethics and privacy to NLP (De Jong et al., 2018; Leidner and Plachouras, 2017; Prabhumoye et al., 2021). Our work follows the precedent established by Blodgett et al. (2020), who used keywords to select papers from the ACL anthology, and then enlarged the sample by following citations to other popular venues (eg AAAI, ICML etc.). Similarly, we also align with Field et al. (2021) who solely focus on papers published at ACL venues but draw on conclusions from NLP papers published elsewhere. Relying on keywords, of course, introduces possibilities for false positives and false negatives.

**Scope of this survey**   In this work, we solely focus on papers published in the ACL anthology to gain a better insight into current trends and popular research approaches for social bias. One limitation of such an approach is that seminal work published at other venues is not reviewed here. In Appendix 7, we provide a table that shows each paper reviewed in this survey.

## 3   Defining Bias

The dictionary definition of bias is "an inclination or prejudice for or against one person or group, especially in a way considered to be unfair" (Stevenson, 2010). Based on this, defining bias involves an interaction among different components: (i) individuals or categories that determine a group, (ii) attitudes towards this group, and (iii) assessment of this attitude in relation to fairness. Bias comes into existence when a specific attitude is formed, which may or may not be fair. For example, in investigating a disease that is more prevalent among women, use of *gender* is a relevant variable and, in itself, does not entail a differential attitude. Once a differential attitude is formed however, bias comes into existence and the attitude may be negative or positive. By definition, if a distinctly positive attitude is formed about one variable (e.g., *gender*), it entails a less favorable attitude about the other *gender* categories. This does not mean that all bias

is necessarily unfair, there are multiple theories and definitions of fairness that are formulated and analyzed in-depth in political philosophy (Lamont and Favor, 2004). The formal principle of equality formulated by Aristotle (Ameriks and Clarke, 2000) states that equals must be treated equally, which is often referred to as 'the fairness ideal', but it is neither a prevailing definition nor a useful one in practice. Without identifying relevant features, such a definition would not prevent categories such as *race* or *wealth* to be used as variables for differential treatment. However, a fairness approach (Lamont and Favor, 2004) based on equal opportunity might require a differential attitude (i.e., bias) towards a certain category in order to 'level the playing field'. For example, if women are routinely given worse performance reviews and lower pay for successfully completing the same tasks as men, then there is differential treatment. Meaning women who are as successful as men cannot have the same opportunities. According to this understanding of fairness, a bias towards women would be fair. It is also worth noting that positive or negative discrimination is distinct from bias. While discrimination is about treatment, bias is about attitude. In other words, bias may lead to discrimination (Mateo and Williams, 2020). In this context, the ethical issue about bias is tied to differential treatment of descriptive categories resulting in unfair outcomes. Identifying and dealing with ethical concerns related to bias, must necessarily involve identifying the descriptive categories and the biases against those categories as well as examining whether the said bias is unfair, according to the relevant definition of fairness.

## 4   Methodology

Following the standard practice mentioned above, we searched the ACL anthology[1] in September 2022 for relevant titles and abstracts using the keywords: *ethical, ethics, fairness, fair, bias, social, society, societal, social good*. For papers, published after September 2022 we manually screened all conference proceedings for the same keywords. One limitation of relying on a keyword search alone is that we might miss any work that refers to a bias directly in the title, for example 'fatphobia detection in online forums'.

---

[1] https://github.com/shauryr/ACL-anthology-corpus

**Filtering Strategy**  A total of *1,437* papers were returned by the search; *523* papers were retained after a manual screening of titles and abstracts. We removed duplicates, as well as work not related to bias and/or ethics in NLP. Then we downloaded full papers, and filtered out papers if: (i) the contribution was a talk, demonstration, abstract or keynote, (ii) "bias" was used in the machine learning sense, or (iii) the paper did not focus on social bias. This process produced a collection of *348* papers.

**Categorization process**  We identify trends in our corpus by empirically determining a set of five categories, where we fully review each paper manually. We make our corpus alongside with its corresponding categories and labels publicly available[2]. We focus on four elements for each paper to identify trends, where we identify language investigated, type of bias analyzed and what kind of paper is introduced and which NLP area it belong to. For the type of paper analysis we utilize the authors description of contributions to split the papers in the following categories:

- **Method:** In this category of papers, the main focus of the work is to contribute a new method, which includes but is not limited to novel ways to measure or mitigate social bias.

- **Analysis** Papers in this category, examine existing datasets, benchmarks, language models, NLP systems or embeddings for bias using social science methods, statistics and mixed methods approaches. For example, authors who have conducted research in this category have explicitly stated that they conduct an analysis or outline a mixed method approach.

- **Surveys and Position papers:** This category of papers includes surveys, guides, tutorials, reviews and position papers.

- **Dataset, Benchmarks or Resources:** Paper in this category propose new datasets, benchmarks, lexicons, challenge sets and often include some preliminary analysis of the new data either collected through crowd-sourcing.

- **Datasets, Benchmarks and Methods:** This is a combination of papers that focus on both introducing a new resource (e.g.: dataset or benchmark) in addition to a new methodology.

---

[2] Link-added-upon-publication

| Year | Papers | Year | Papers |
|------|--------|------|--------|
| 2010-2016 | 11 | 2020 | 68 |
| 2017 | 16 | 2021 | 96 |
| 2018 | 10 | 2022 | 79 |
| 2019 | 46 | 2023 | 22 |
| **Totals** | **83** | **Totals** | **265** |

Table 1: 76,15% of the 348 papers are from 2020-2023.

## 5  Empirical Findings

The 348 papers were published between 2010 and 2023. Table 1 shows that there has been considerable growth in interest in bias research.

**Type of Paper**  Based on the criteria outlined above, we have found that the majority of papers introduced are *Method* papers (57.75%), followed by *Datasets and Benchmarks* (20.40%), *Surveys* (15.22%), *Analysis* (4.31%) and combined *Datasets, Benchmarks and Methods* (2.29%). We take a closer look the majority category *Methods*, where we assigned *Method* papers into different categories based on the contribution of the paper (see Table 2).

**Methods**  We distinguish between bias detection and measurement, as detection does not necessarily measure or remove any kind of bias. We found similarly to (Blodgett et al., 2020) that many papers propose a combination of techniques, hence why we have decided to merge such approaches into the category *'Measurement and Mitigation'*. Many works in Debiasing and Mitigation apply or extend methods such as WEAT (Caliskan et al., 2017) or HARD DB (Bolukbasi et al., 2016) to a specific benchmark or dataset. One negative side effect of this could be that for example in *gender bias* Measurement/Mitigation, there is no evidence that HardDB can or should be applied to languages with grammatical gender (Sun et al., 2019). There has also been criticism of removing bias (Caliskan et al., 2017), where removing bias (i) only changes how an application or algorithm understands the world but not how it applies the knowledge gained from understanding ('fairness through blindness'), (ii) could harm meaning and accuracy and (iii) bias can (unintentionally) be introduced through other avenues during the design process. Therefore, simply removing bias is not enough (Chandrabose et al., 2021) and developing new methods requires consistent reflection as bias in NLP systems is never fully inescapable (Waseem et al., 2021). Work by (Hovy, 2015; Kiritchenko and Mohammad, 2018) has found that the some

information bias mitigation techniques can be beneficial in improving performance in downstream tasks. The methods in the *Miscellaneous* category take a different approach to working on bias. For example (Fisher et al., 2020) in including bias sensitive attributes by defining a whitelist of triples for uncontroversial cases. Arguably, one drawback here is that it is up to the person whitelisting to decide what is not controversial and what is. Similarly, (Touileb et al., 2021) and (Wang et al., 2017) use *gender* and *linguistic bias* respectively to improve classification results. Touileb et al. (2021) first show how female critics disproportionately give lower ratings to female authors, where removing metadata may have the opposite effect in that it does not help traditionally underrepresented groups in a specific domain. At the same time, many of the methods looking at measuring or removing bias complicate the data and tasks at hand and can lead to the development of systems that are not reliable when used in a more complex context (Talat et al., 2022b). This also applies to the predominant use of intrinsic metrics in bias measurement. These metrics may shed more light on how much bias exists in a dataset/LM, but does not necessarily correlate with performance on downstream tasks and therefore does not show the true harm of bias (Orgad and Belinkov, 2022). Thus, we may run the risk of developing methods for each new dataset or benchmark and missing out on crucial information that shows how bias affects different downstream tasks in different ways. However, documenting and measuring bias in a systematic way is crucial to understanding what harms can be caused in real life situations, so that preventive methods can be developed (Dev et al., 2021b). Current approaches in mitigation and/or measurement methods are evaluated on a variety of NLP areas, including Language Models (35.74%), Classification (22.85%) , NLG (14.76%) and NLI (3.33%). It is unsurprising that much attention has been paid to embedding representations that are trained on large amounts of text (Kiritchenko and Mohammad, 2018; Mayfield et al., 2019; Talat et al., 2022b). This has the benefit of bias methods being more widely applicable, but it also means that there are distinct limitations when a method is tied to a specific architecture rather than the task/benchmark itself. It means bias measures are no longer comparable in relation to other benchmarks and bias can be introduced at any stage of an NLP system design as it de-

| Type | Papers | % |
|---|---|---|
| Measurement | 98 | 46.66 |
| Debiasing / Mitigation | 52 | 24.76 |
| Combinations of above | 32 | 15.23 |
| Detection | 19 | 9.04 |
| Generation | 5 | 2.38 |
| Miscellaneous | 4 | 1.9 |
| **Total** | 210 | 100.0 |

Table 2: Empirical taxonomy of methods.

pends on where and how the final LLM is applied and to which community (Talat et al., 2022b). An important trade-off to consider is the balance between generalizable and context-sensitive methods to measure bias in downstream tasks. There are also other areas that have done work on bias but are not represented as well in this survey, which include but are not limited to Speech Recognition (Kwako et al., 2022; Savoldi et al., 2022), Multi-Modal NLP (Chen et al., 2020a; Srinivasan and Bisk, 2022a) and Information Extraction (Li et al., 2022b; Sun and Peng, 2021).

**Social biases** In Figure 1 we show the types of biases investigated, where we include all social biases that occur more than once (see diagonal of the matri). Furthermore there are a small number of biases that only occur once and are not shown in the table, such as bias transfer hypothesis (Steed et al., 2022) or dialect bias (Tatman, 2017b). Furthermore, we included the category *multiple social biases*, where the paper does not explicitly list or describe the specific type of bias investigated (Ghosh et al., 2021; Ramponi and Tonelli, 2022; Mireshghallah and Berg-Kirkpatrick, 2021; Loukina et al., 2019). There also is the *intersectional bias* category, which shows how different elements of a person's identity (e.g., gender, race and age) can either be a benefit or disadvantage and lead to compounded discrimination (Crenshaw, 2013). For example, recent work by Lalor et al. (2022) benchmarks a variety of NLP models on different downstream tasks for its performance on intersectional biases and (Câmara et al., 2022) introduce a framework for unisectional and intersectional analysis of sentiment analysis in a multilingual setting. Few works focus on biases other than *gender*, where (Davidson et al., 2019; Sap et al., 2019; Manzini et al., 2019) look at *racial bias* and (Hutchinson and Mitchell, 2019; Herold et al., 2022) investigate *disability bias*. Noticeable is that some biases are not investigated on their own, such as *age*, *religion*, *sexuality* or *profession*. We included *political/media bias* in this analysis, if the paper also looks at social bias. For example, debiasing claims that include

attitudes towards a group (e.g., *sexuality*). However, this type of work does not explicitly mention social biases when attitudes or characteristics of the targeted group are only implied. Dayanık and Padó (2020) looks media bias on a immigration dataset (MARDY) but does not mention implied social biases (e.g., *nationality* or *ethnicity*). The most frequently combined social biases are *gender* and *race*.The most frequently combined social biases are *gender* and *race*. In Table 3 we compute residuals between observed joints and predictions on margin, where highlighted in *green* are highly saturated areas and shades of *red* show less saturated areas.

**Languages**   We show the languages used in each type of paper, excluding *Surveys and Position papers* in Figure 2. There are a total of 34 languages, however we leave out any languages that only occur once in the visualization. There are 11 languages not visualized, including *Farsi*, *Urdu*, *Wolof*, *Bengali*, *Armenian*, *Bengali*, *Inuktitut*, *Ukrainian*, *Hungarian*, *Indonesian* and *Lithuanian*. English, German, Spanish and Chinese are most commonly used either on their own or in combination with each other. The majority of all papers focus on a single language at a time. Furthermore, the vast majority of LLMs are monolingual and do not encode the cultural variety that naturally occurs within one language, for example non-standard English varieties (Talat et al., 2022b). Therefore it is important to not only document the type of bias investigated, but also contextualize bias within a language's cultural context, understanding of said bias and document the language itself (*Bender Rule* (Bender, 2011)). Based on this collection, bias research is heavily biased towards western and Anglo-centric notions of bias and very few works focus on non-English benchmarks (Talat et al., 2022b; Hovy and Spruit, 2016). This proves extremely problematic when English benchmarks are automatically translated, but many of the biases do not hold true in non-Western cultural contexts. For example, gendered professions do not necessarily translate across every language or culture (Talat et al., 2022b) and many NLP systems trained on written English (e.g., Penn Tree Bank) do not perform well on non-standard English (Mayfield et al., 2019). From Table 4 we can see the residuals between observed joints showing a clear over-saturation (*green*) for specific combinations of languages (e.g: *English* and *German* or *English* and *Spanish*).

# 6   Discussion

**Datasets and Benchmarks**   Previous work (Hovy and Spruit, 2016; Hovy, 2018; Talat et al., 2022b) outlined a number of reasons that may explain why there are so many papers focusing on the same datasets, benchmarks, biases, and languages. In the following section, we highlight some of the elements that may explain why there is an uneven distribution of work and resources.

- **Experimental setup** The majority of of work in bias research has focused on using intrinsic bias measurements (bias in internal model representations) and little attention has been paid to extrinsic metrics (Orgad and Belinkov, 2022; Delobelle et al., 2022). A very real consequence of this is that much work does not appropriately describe, contextualize and identify the potential harms that bias has in real-world scenarios (Blodgett et al., 2020). Also bias is inadvertently introduced in intrinsic metrics, where lexicons used to measure bias in one dataset produces very different results in another (Antoniak and Mimno, 2021). Similarly, (Goldfarb-Tarrant et al., 2021) have found that there is no correlation between intrinsic and extrinsic metrics. Another key problem is that often newly proposed datasets are linked to specific metrics, which makes it hard to draw conclusions from individual case studies as many results are not generalizable (Orgad and Belinkov, 2022). Another element impacting metrics is the composition of test data, where (Orgad and Belinkov, 2022) found that test sets often don't contain balanced examples. However, most metrics are defined over a whole dataset and are therefore sensitive to its composition, which may lead to variability in results. Both metric choice and dataset composition can significantly change the results and conclusions drawn from a downstream task or dataset (Akyürek et al., 2022a). Therefore, it is important to (i) provide motivation for including / excluding a particular metric and describe how it impacts downstream performance, (ii) compare a metric across a variety of datasets and (iii) compare many metrics across individual datasets.

- **Funding** There are unintended consequences of research that can be traced to how research projects are funded (e.g., governments or mil-

| | gender | race | religion | age | sentiment | profession | media | nationality | political | multiple social | annotator | sexuality | disability | ethnicity | physical appearance | dialect | human | intersectional | |
|---|---|---|---|---|---|---|---|---|---|---|---|---|---|---|---|---|---|---|---|
| | | 19.3 | 8.2 | 8.8 | -8.5 | 5.8 | -8.6 | 3.0 | -6.5 | -7.5 | -6.5 | 5.1 | 1.7 | 3.8 | 3.0 | 1.1 | -2.3 | -1.7 | gender |
| | 19.3 | | 12.2 | 12.4 | -1.0 | 6.8 | -3.0 | 4.2 | -1.6 | -2.6 | -2.6 | 8.6 | 3.8 | 4.2 | 4.6 | 0.0 | -0.8 | -0.6 | race |
| | 8.2 | 12.2 | | 4.4 | -1.4 | 7.9 | -1.0 | 7.0 | -0.9 | -0.9 | -0.9 | 4.2 | 4.2 | 2.4 | 5.5 | 0.7 | -0.3 | -0.2 | religion |
| | 8.8 | 12.4 | 4.4 | | -1.3 | 3.9 | -1.0 | 6.1 | 0.1 | -0.9 | 0.1 | 4.2 | 5.3 | 4.4 | 5.5 | 0.7 | -0.3 | -0.2 | age |
| | -8.5 | -1.0 | -1.4 | -1.3 | | 0.1 | 0.1 | 0.2 | -0.7 | -0.7 | -0.7 | -0.7 | -0.6 | -0.5 | -0.4 | -0.3 | -0.2 | 0.8 | sentiment |
| | 5.8 | 6.8 | 7.9 | 3.9 | 0.1 | | -0.7 | 5.4 | -0.6 | -0.6 | -0.6 | 3.4 | 4.5 | 0.6 | 4.7 | -0.2 | -0.2 | -0.1 | profession |
| | -8.6 | -3.0 | -1.0 | -1.0 | 0.1 | -0.7 | | -0.6 | -0.6 | -0.6 | -0.6 | -0.5 | -0.5 | -0.4 | -0.3 | -0.2 | -0.2 | -0.1 | media |
| | 3.0 | 4.2 | 7.0 | 6.1 | 0.2 | 5.4 | -0.6 | | -0.5 | -0.5 | -0.5 | 3.5 | 5.6 | 3.6 | 4.7 | -0.2 | -0.2 | -0.1 | nationality |
| | -6.5 | -1.6 | -0.9 | 0.1 | -0.7 | -0.6 | -0.6 | -0.5 | | -0.5 | 0.5 | 0.6 | 0.6 | 0.7 | -0.3 | -0.2 | -0.1 | -0.1 | political |
| | -7.5 | -2.6 | -0.9 | -0.9 | -0.7 | -0.6 | -0.6 | -0.5 | -0.5 | | -0.5 | -0.4 | -0.4 | -0.3 | -0.3 | -0.2 | -0.1 | -0.1 | multiplesocial |
| | -6.5 | -2.6 | -0.9 | 0.1 | -0.7 | -0.6 | -0.6 | -0.5 | 0.5 | -0.5 | | -0.4 | -0.4 | -0.3 | -0.3 | -0.2 | -0.1 | -0.1 | annotator |
| | 5.1 | 8.6 | 4.2 | 4.2 | -0.7 | 3.4 | -0.5 | 3.5 | 0.6 | -0.4 | -0.4 | | 4.6 | 0.7 | 3.8 | -0.2 | -0.1 | -0.1 | sexuality |
| | 1.7 | 3.8 | 4.2 | 5.3 | -0.6 | 4.5 | -0.5 | 5.6 | 0.6 | -0.4 | -0.4 | 4.6 | | 1.7 | 4.8 | -0.2 | -0.1 | -0.1 | disability |
| | 3.8 | 4.2 | 2.4 | 4.4 | -0.5 | 0.6 | -0.4 | 3.6 | 0.7 | -0.3 | -0.3 | 0.7 | 1.7 | | 0.8 | -0.1 | -0.1 | -0.1 | ethnicity |
| | 3.0 | 4.6 | 5.5 | 5.5 | -0.4 | 4.7 | -0.3 | 4.7 | -0.3 | -0.3 | -0.3 | 3.8 | 4.8 | 0.8 | | -0.1 | -0.1 | -0.1 | physicalappearance |
| | 1.1 | 0.0 | 0.7 | 0.7 | -0.3 | -0.2 | -0.2 | -0.2 | -0.2 | -0.2 | -0.2 | -0.2 | -0.2 | -0.1 | -0.1 | | -0.1 | -0.0 | dialect |
| | -2.3 | -0.8 | -0.3 | -0.3 | -0.2 | -0.2 | -0.2 | -0.2 | -0.1 | -0.1 | -0.1 | -0.1 | -0.1 | -0.1 | -0.1 | -0.1 | | -0.0 | human |
| | -1.7 | -0.6 | -0.2 | -0.2 | 0.8 | -0.1 | -0.1 | -0.1 | -0.1 | -0.1 | -0.1 | -0.1 | -0.1 | -0.1 | -0.1 | -0.0 | -0.0 | | intersectional |

Table 3: Observed joints: number of papers with combinations of languages (ISO 639). biases

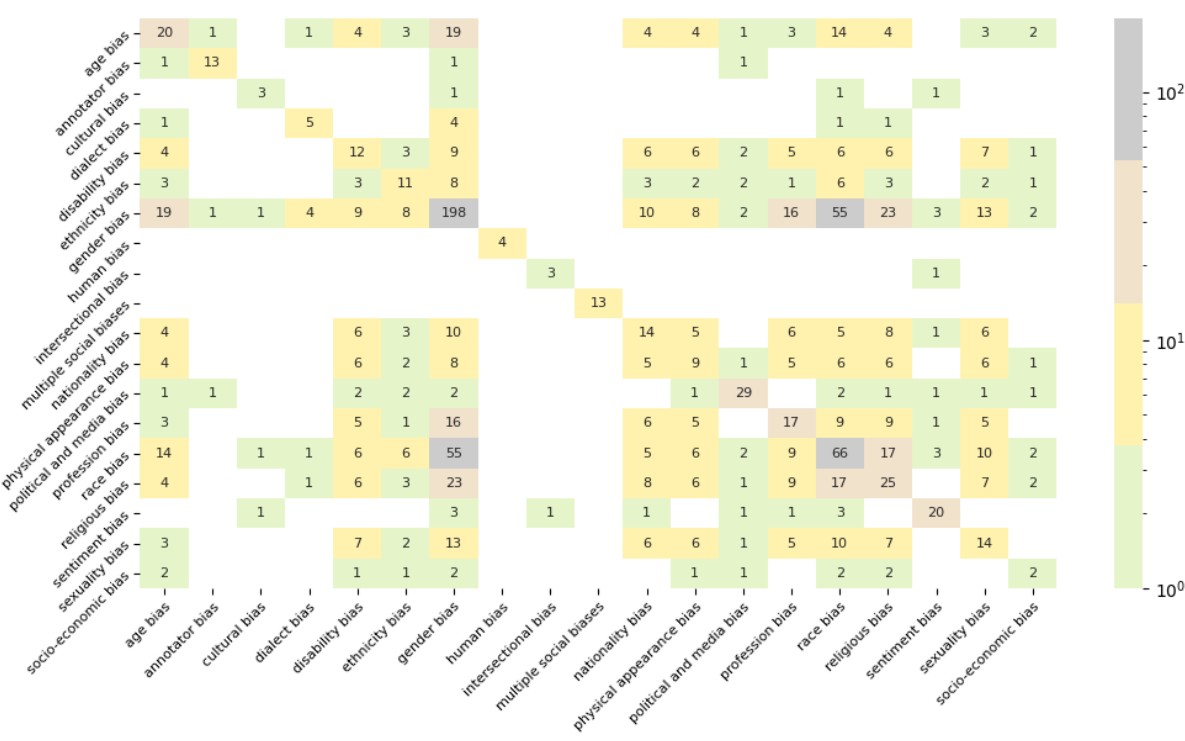

Figure 1: A log-scaled heatmap showing the type and frequency of social biases.

itary interests), where researchers should be aware that their work has broader impact and can be abused (Hovy and Spruit, 2016).

- **Availability and Overexposure** We have found that a small number of papers introduce new datasets or benchmarks (see 5). Creating and curating new datasets as well as benchmarks are often a time-consuming, expensive and long process, where it is oftentimes easier to utilize existing resources to try out new methods (Hovy, 2018). Similarly there is the phenomenon of *topic overexposure*, where there are waves of seemingly 'popular' re-

| | en | de | sp | zh | fr | it | ru | tr | pl | nl | ar | pt | da | te | sv | no | ja | hi | sk | ko | he | |
|----|----|----|----|----|----|----|----|----|----|----|----|----|----|----|----|----|----|----|----|----|----|----|
| en | | 0.3 | 2.1 | 2.9 | -4.2 | 1.6 | 0.8 | -1.2 | 1.5 | -0.7 | -1.7 | -0.7 | 0.0 | -2.0 | -2.2 | -1.2 | -1.2 | -1.2 | -2.2 | -0.5 | 0.5 | en |
| de | 0.3 | | 2.7 | 3.7 | 1.8 | 1.8 | 0.8 | 0.9 | 0.9 | 1.9 | -0.1 | 2.9 | 0.9 | -0.1 | -0.0 | -0.0 | -0.0 | -0.0 | -0.0 | 1.0 | 1.0 | de |
| sp | 2.1 | 2.7 | | 8.7 | 5.0 | 7.0 | 4.2 | 4.5 | 4.6 | 1.7 | 1.7 | 2.7 | 2.7 | 0.7 | -0.2 | -0.2 | -0.2 | 1.8 | -0.2 | 0.9 | 1.9 | sp |
| zh | 2.9 | 3.7 | 8.7 | | 3.2 | 6.2 | 5.4 | 3.6 | 3.7 | 1.7 | -0.3 | 3.7 | 3.8 | 0.8 | -0.2 | -0.2 | -0.2 | 0.8 | -0.2 | 0.9 | 1.9 | zh |
| fr | -4.2 | 1.8 | 5.0 | 3.2 | | 1.4 | 0.5 | 1.7 | 1.7 | 0.8 | -0.2 | 2.8 | 1.8 | 0.8 | -0.1 | 0.9 | -0.1 | 1.9 | -0.1 | 0.9 | -0.1 | fr |
| it | 1.6 | 1.8 | 7.0 | 6.2 | 1.4 | | 6.6 | 1.7 | -0.2 | -0.2 | 0.8 | 1.8 | 0.8 | 0.8 | -0.1 | -0.1 | 0.9 | -0.1 | -0.1 | -0.1 | 0.9 | it |
| ru | 0.8 | 0.8 | 4.2 | 5.4 | 0.5 | 6.6 | | 1.8 | 0.8 | 0.8 | 0.8 | 0.8 | 1.9 | 0.9 | -0.1 | 0.9 | -0.1 | -0.1 | -0.1 | -0.1 | 0.9 | ru |
| tr | -1.2 | 0.9 | 4.5 | 3.6 | 1.7 | 1.7 | 1.8 | | 1.9 | -0.1 | -0.1 | 1.9 | 0.9 | 0.9 | -0.1 | -0.1 | -0.1 | 0.9 | -0.1 | -0.0 | 1.0 | tr |
| pl | 1.5 | 0.9 | 4.6 | 3.7 | 1.7 | -0.2 | 0.8 | 1.9 | | 1.9 | -0.1 | -0.1 | 0.9 | -0.1 | -0.1 | -0.1 | -0.1 | -0.1 | -0.1 | -0.0 | 1.0 | pl |
| nl | -0.7 | 1.9 | 1.7 | 1.7 | 0.8 | -0.2 | 0.8 | -0.1 | 1.9 | | -0.1 | -0.1 | 1.9 | -0.1 | -0.0 | -0.0 | -0.0 | -0.0 | -0.0 | 1.0 | -0.0 | nl |
| ar | -1.7 | -0.1 | 1.7 | -0.3 | -0.2 | 0.8 | 0.8 | -0.1 | -0.1 | -0.1 | | -0.1 | -0.1 | -0.1 | -0.0 | 1.0 | 1.0 | -0.0 | -0.0 | -0.0 | 1.0 | ar |
| pt | -0.7 | 2.9 | 2.7 | 3.7 | 2.8 | 1.8 | 0.8 | 1.9 | -0.1 | -0.1 | -0.1 | | 0.9 | -0.1 | -0.0 | -0.0 | -0.0 | 1.0 | -0.0 | -0.0 | 1.0 | pt |
| da | 0.0 | 0.9 | 2.7 | 3.8 | 1.8 | 0.8 | 1.9 | 0.9 | 0.9 | 1.9 | -0.1 | 0.9 | | 1.0 | -0.0 | -0.0 | -0.0 | 1.0 | -0.0 | -0.0 | -0.0 | da |
| te | -2.0 | -0.1 | 0.7 | 0.8 | 0.8 | 0.8 | 0.9 | 0.9 | -0.1 | -0.1 | -0.1 | -0.1 | 1.0 | | -0.0 | 1.0 | -0.0 | -0.0 | -0.0 | -0.0 | -0.0 | te |
| sv | -2.2 | -0.0 | -0.2 | -0.2 | -0.1 | -0.1 | -0.1 | -0.1 | -0.1 | -0.0 | -0.0 | -0.0 | -0.0 | -0.0 | | -0.0 | -0.0 | -0.0 | -0.0 | -0.0 | -0.0 | sv |
| no | -1.2 | -0.0 | -0.2 | -0.2 | 0.9 | -0.1 | 0.9 | -0.1 | -0.1 | -0.0 | 1.0 | -0.0 | -0.0 | 1.0 | -0.0 | | -0.0 | -0.0 | -0.0 | -0.0 | -0.0 | no |
| ja | -1.2 | -0.0 | -0.2 | -0.2 | -0.1 | 0.9 | -0.1 | -0.1 | -0.1 | -0.0 | 1.0 | -0.0 | -0.0 | -0.0 | -0.0 | -0.0 | | -0.0 | -0.0 | -0.0 | -0.0 | ja |
| hi | -1.2 | -0.0 | 1.8 | 0.8 | 1.9 | -0.1 | -0.1 | 0.9 | -0.1 | -0.0 | -0.0 | 1.0 | 1.0 | -0.0 | -0.0 | -0.0 | -0.0 | | -0.0 | -0.0 | -0.0 | hi |
| sk | -2.2 | -0.0 | -0.2 | -0.2 | -0.1 | -0.1 | -0.1 | -0.1 | -0.1 | -0.0 | -0.0 | -0.0 | -0.0 | -0.0 | -0.0 | -0.0 | -0.0 | -0.0 | | -0.0 | -0.0 | sk |
| ko | -0.5 | 1.0 | 0.9 | 0.9 | 0.9 | -0.1 | -0.1 | -0.0 | -0.0 | 1.0 | -0.0 | -0.0 | -0.0 | -0.0 | -0.0 | -0.0 | -0.0 | -0.0 | -0.0 | | -0.0 | ko |
| he | 0.5 | 1.0 | 1.9 | 1.9 | -0.1 | 0.9 | 0.9 | 1.0 | 1.0 | -0.0 | 1.0 | 1.0 | -0.0 | -0.0 | -0.0 | -0.0 | -0.0 | -0.0 | -0.0 | -0.0 | | he |

Table 4: Residuals between observed joints and predictions based on margins.

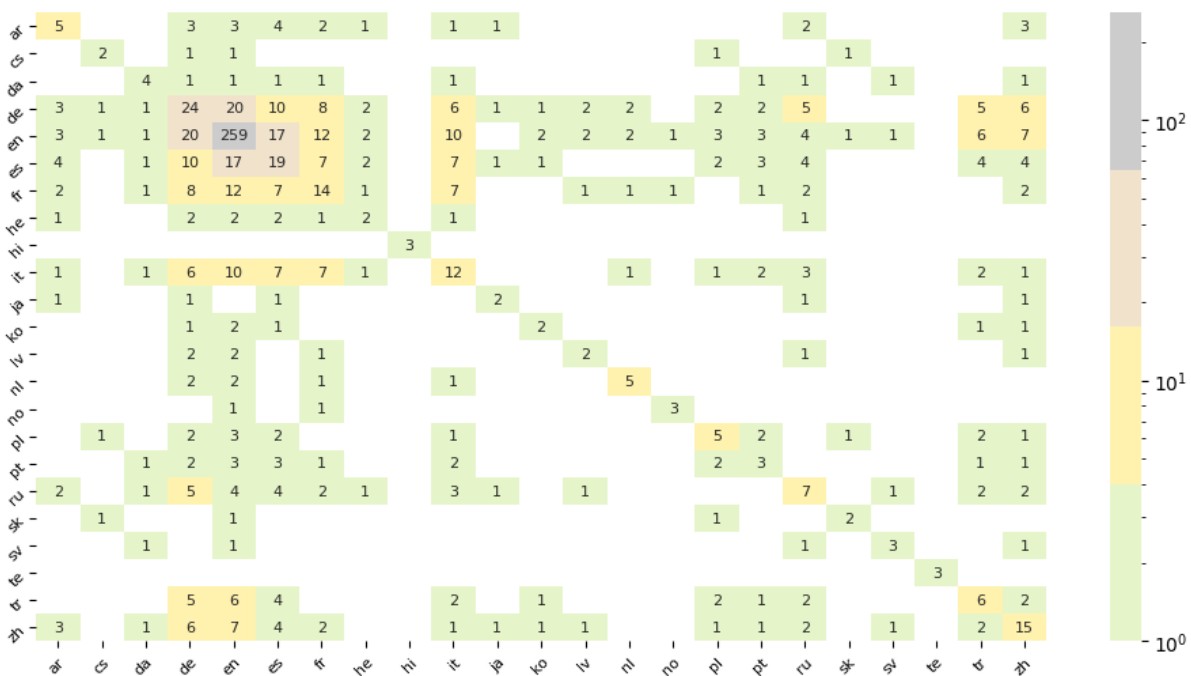

Figure 2: A log-scaled heatmap showing the different languages (ISO-639-1) and their frequency of occurrence.

search topics that eventually go out of fashion. This is based on availability heuristic, if people recall a certain event or have knowledge about certain things then it must be important (Hovy and Spruit, 2016).

**Bias** We have found a limited focus on specific social biases, where possible causes are rooted in (i) the data that encodes bias by default (Chandrabose et al., 2021), where already available data determines what kind of bias we focus on, (ii) machine learning breakthroughs in NLP has enabled 'streetlamp science' and we focus on tasks that can be solved (Hovy, 2018) and (iii) lack of awareness. This has the consequence that difficult tasks are not being tackled and bias remains present in NLP tools. Therefore it is key to raise awareness (Baeza-Yates, 2018), understand and measure what kind of bias has influence on NLP models and work towards developing solutions that are equitable. Here,

we (i) showcase challenges in three frequently researched social biases that have been identified through this survey and (ii) point out opportunities for the future with the aim to raise awareness.

- **Gender bias** There is a strong emphasis on a binary understanding of gender (Schnoebelen, 2017; Orgad and Belinkov, 2022) and most task have been reduced to a masculine/feminine dichotomy (Savoldi et al., 2021). Initially, this may be perceived as useful to enable research, however it does not capture the reality of the society and world we live in today. For example, in the USA alone over 1.4 million people identify as transgender (Larson, 2017) and 1.2 million identify as non-binary (Williams Institute, 2021). At the same time, there is not only a common misconception that gender and sex are the same (Larson, 2017), but also that sexuality is somehow indicative of either gender or sex. Sexuality refers to a person's attraction to a sex or gender, but is not a marker of gender/sex itself (Baum and Westheimer, 2015). Therefore, one can not use sex or gender as a predictor or precursor to assuming a person's sexuality. Talat et al. (2022b) argue that characteristics like sexuality are usually not observable, which can lead to a reliance on hegemonic stereotypes and unnatural language in bias evaluation benchmarks. This leaves plenty of opportunity to start conversations around developing new datasets, benchmarks and methods that are more inclusive and reflect the world we live in (Savoldi et al., 2021; Orgad and Belinkov, 2022).

- **Race bias** Related work by (Field et al., 2021) provides an excellent overview of the state of the art of race bias research in NLP. In their survey they identify that there are very few datasets and benchmarks and that oftentimes a narrow view of race and racial identity are perpetuated. Additionally, researchers often doesn't explicitly state if they are focusing on racial bias through downstream tasks such as abusive language detection. Subsequently, currently deployed hate speech or toxicity classifiers mislabel language predominantly used in the African American community as toxic or hate speech when it is not (Dixon et al., 2018; Xia et al., 2020).

Finally, this survey mentions a number of social biases that have been mentioned such as *religious*, *age* and *disability* with few papers in Figure 1. It is outside of the scope of this work to address each social bias individually, but we would like to point out that there is a lack of relevant benchmarks, datasets and surveys to make substantial progress in these areas and understand the unique challenges each community faces (individually and at an intersectional-level). Most importantly, we would like to emphasize that this type of future work needs to be deeply grounded in interdisciplinary research and led by diverse teams that connect and engage with relevant communities.

**Interdisciplinary research** The relationship between language and social hierarchies is far more complex than what current techniques can capture. Therefore new methods need to be grounded in relevant literature outside of NLP (Blodgett et al., 2020), because social bias is a complex issue (Sun et al., 2019). Whilst NLP researchers may be committed to using ethical approaches, they may not necessarily have the required ethical and legal knowledge to do so (Santy et al., 2021). This makes it incredibly important to foster collaboration between disciplines to ensure that historical inequalities and biases are taken into consideration when building new algorithms or systems (Caliskan et al., 2017).

**Diversity** Given the real-life impact of NLP systems and research on people, there is not just a need for diversity in experts working on such systems (Caliskan et al., 2017), but also a need for practitioners and researchers to engage with the affected communities and stakeholders (Blodgett et al., 2020; Fortuna et al., 2021). Therefore, we need to recognize the implicit bias of the people working on different NLP systems and sense-check at different stages how this bias may be reflected in collected data, new benchmarks or models (Savoldi et al., 2021; Hovy and Spruit, 2016). We also need to acknowledge the lack of diversity in teams working on NLP (Schluter, 2018; Savoldi et al., 2021) and work towards more inclusive teams that represent a wide variety of backgrounds and lived experiences (Field et al., 2021). Otherwise, NLP systems continue to represent majorities and we risk the further oppression of already disadvantaged communities (Talat et al., 2022b; Schnoebelen, 2017).

## Limitations and Ethics Statement

In this paper, we surveyed a collection of papers and identified continued challenges in social bias research. We have created this collection based on a keyword search and outlined how this may not fully reflect all literature on social bias existing in the ACL anthology or other venues. We only used open-access papers in this collection and no human participants were involved in this work. Traditionally, social biases have been investigated in fields such as social sciences, law, or psychology which we have not discussed here. Furthermore, we do not give an analysis of algorithmic or dataset biases (e.g., machine learning, data mining or otherwise) or provided an in-depth review of technical contributions in computational social biases. We are also limited by the resource of the reviewed papers, where substantial contributions to the field have been made outside of ACL venues. Finally, we would like to point out that opportunities and recommendations for future bias research as proposed in section 6 should be considered from a euro- and/or anglo-centric perspective. There may be a variation depending on the social context, country or culture that works on a specific bias problem.

## Acknowledgements

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

# 7 Appendix

| Type | Papers |
| --- | --- |
| Survey/Position Paper | Drugan and Babych (2010), Masthoff (2011), Escartín et al. (2016), Hovy and Spruit (2016), Leidner and Plachouras (2017), et al. (2017), Schnoebelen (2017), Larson (2017), De Jong et a (2018), Mayfield et al. (2019), Sun et al. (2019), Blodgett et al. (2021), Bender et al. (2020), Schoch et al. (2020), Savoldi et a (2021), Mishra et al. (2021), Czarnowska et al. (2021), Dough Sheng et al. (2021), Hovy and Yang (2021), Santy et al. (2021), (2022b), Chandrabose et al. (2021), Vecchi et al. (2021), Talat et a Balkir et al. (2022), Orgad and Belinkov (2022), Akyürek et al (2022), Mohammad (2022a), Orr et al. (2022), Benotti and Blac et al. (2023), Benotti et al. (2023), Ramesh et al. (2023), Rios et |
| Dataset, Benchmark | Yano et al. (2010), Flekova et al. (2016), Braun et al. (2016) Ka Rudinger et al. (2018), Zhao et al. (2018), Kiritchenko and Moh (2019) , Gaut et al. (2020), Friedman et al. (2019), Hitti et al. ( (2019), Fan et al. (2019), Liu et al. (2020a), Sap et al. (2020), Lou et al. (2021), Lim et al. (2020), Shahid et al. (2020), Lazarido Uthus (2020), Gala et al. (2020), Dinan et al. (2020b), Meyer e (2020b), Gangula et al. (2019), Zueva et al. (2020), Nangia et al. Blodgett et al. (2021), Spinde et al. (2021), Zhong et al. (2021), Søgaard (2021), Zhao et al. (2021), Aksenov et al. (2021), Kam Renduchintala et al. (2021), Barikeri et al. (2021), Miłkowski et a et al. (2022), Ziems et al. (2022), Zhou et al. (2022), Akyürek e (2022), Bansal et al. (2022), Smith et al. (2022), Qian et al. (2022 et al. (2023), Madnani et al. (2017), Kumar and Bhotia (2020) (2021), Zad et al. (2021), Nozza et al. (2022), Měchura (2022) |
| Methods | Gonen and Goldberg (2019), Wevers (2019), Hovy et al. (2020 Nissim et al. (2020), Wang et al. (2021d), Wolfe and Caliskan (2 et al. (2021), Herold et al. (2022) , Bertsch et al. (2022), Ramp Cao et al. (2022a), Alshahrani et al. (2022) Daems and Hackenbu et al. (2015) , Terkik et al. (2016), Tatman (2017b), Wang et al Tatman (2017a), Zhao et al. (2017), Long et al. (2018), Baly e Kloppenburg (2019), Färber et al. (2019), Zhong et al. (2019), T and Ungar (2019), Gangula et al. (2019), Karve et al. (2019), and Najafian (2019), Chaloner and Maldonado (2019) , Qian et (2019) , Kaneko and Bollegala (2019), Huang et al. (2020a), Ba Dinan et al. (2020a), Bordia and Bowman (2019) , Manzini et al. , Schwertmann et al. (2023), Pedersen et al. (2023), Bauer et al. ( Kumar et al. (2023) , Touileb et al. (2023), Parmar et al. (2022), et al. (2023), Alemany et al. (2023), Martinková et al. (2023), F (2022), Cao et al. (2022b), Sap et al. (2020) , Xia et al. (2020) (2022) , Orgad et al. (2022), Sen et al. (2022), Gupta et al. (202 Gira et al. (2022), Guo et al. (2022), Wang et al. (2022), An et a (2022), Borchers et al. (2022), Touileb et al. (2022), Jentzsch and et al. (2022), Tal et al. (2022), Li et al. (2022b) , Limisiewicz an et al. (2022) , Ahn et al. (2022), Joniak and Aizawa (2022), Chen (2022), Du et al. (2022) , Agrawal et al. (2022), Sesari et al. (202 Steed et al. (2022), Schick et al. (2021), Zhou et al. (2021), Kane et al. (2021) , Lucy and Bamman (2021), Jin et al. (2021a), L (2021b), Silva et al. (2021), Sun and Peng (2021), Wang et al. (2 , Han et al. (2021), Sen et al. (2021), Ramesh et al. (2021), Sub and Oh (2021) , Davidson et al. (2019), May et al. (2019), Sap Piper (2020), Guo et al. (2020) , Vu et al. (2020), Spliethöver a Munro and Morrison (2020), Zhao and Chang (2020), Schmahl (2020), Chen et al. (2020b) , Mulsa and Spanakis (2020), Shin Cotterell (2020), Liang et al. (2020), Wich et al. (2020a) , Jia et a (2020) , Dev et al. (2021a), Kumar et al. (2020), Field and Tsvetk (2020a), Stafanovičs et al. (2020) , Li et al. (2020), Ma et al. (2 Al Kuwatly et al. (2020), Bartl et al. (2020) , Liu et al. (2020b), H , Sheng et al. (2020), Gaci et al. (2022), Hutchinson et al. (2020) et al. (2021e) , Van Der Wal et al. (2022), Hansen and Søgaard (2 Azarpanah and Farhadloo (2021), Jiao and Luo (2021) , Touileb , Ciora et al. (2021), Murayama et al. (2021), Gaido et al. (2021 (2022), Dawkins (2021a) , Gillis (2021), Mireshghallah and B Wang et al. (2021a), Malik et al. (2022), Subramanian et al. (202 Garimella et al. (2021) , Ghosh et al. (2021), Mehrabi et al. (202 |
| Both above | Goldfarb-Tarrant et al. (2021), Janghorbani and De Melo (2022 Troles and Schmid (2021), Liu et al. (2022a), Névéol et al. (202 |
| Analysis | Herzig et al. (2011), Rudinger et al. (2017), Lauscher and Glava Vanderlyn et al. (2021) , Falenska and Çetinoğlu (2021), Sap et a (2021), Bhat et al. (2021), Haroutunian (2022) , Qian et al. (201 |