# OpenReview forum: "Are we biased on bias? Characterizing social bias research in the ACL community "
_EMNLP/2023/Conference — Submitted to EMNLP 2023_

### Official Review · Reviewer_2HNJ · 2023-08-02

**Paper Topic And Main Contributions:** 1. The paper analyzes a corpus of 348…
**Soundness:** 3

**Excitement:**

2: Mediocre: This paper makes marginal contributions (vs non-contemporaneous work), so I would rather not see it in the conference.

**Questions For The Authors:**

1. The analysis focuses on papers from the ACL anthology. Could you discuss any potential limitations or biases that might introduce, compared to a cross-venue survey?
2. Could you elaborate on how you envision your categorization and findings directly informing actionable next steps for the community?

**Reasons To Accept:**

1. Benchmarking: The corpus provides a benchmark to track the evolution of work on bias in NLP over time. This can reveal progress and persistent gaps.
2. By analyzing trends, the paper highlights under-studied biases, languages, and methods. This points the community towards fruitful future work. It also draws attention to the need for greater diversity and community engagement in bias research.

**Reasons To Reject:**

1. The main contribution is descriptive and remains empirical - categorizing 348 existing papers on social bias. There are no new datasets or algorithms presented. Beyond characterizing current trends, the paper does not provide details on how the analysis could inform concrete next steps. The future recommendations are broad.
2. While identifying gaps can point to open research questions, the paper does not provide technical solutions to fill these gaps. Furthermore, there is no discussion of potential applications that could directly build on the literature survey and analysis.

**Reproducibility:**

4: Could mostly reproduce the results, but there may be some variation because of sample variance or minor variations in their interpretation of the protocol or method.

**Reviewer Confidence:**

5: Positive that my evaluation is correct. I read the paper very carefully and I am very familiar with related work.

---

> ### Author Rebuttal · Authors · 2023-08-28
>
> Firstly, we would like to thank the reviewers for their kind and considered answers that will ultimately help us to improve our work. We have tried our best to answer the questions asked from all reviewers and cross-post them for all to see.
>
> What do you consider the main theoretical basis of your definition of bias?
> From the perspective of ethics and philosophy, bias definition has to be constructed in two steps: (1) what does "bias" mean and (2) what is an "unfair" bias. We take the dry definition of bias (which in itself does not necessarily imply an unfairness or negativity) from dictionary. The dictionary definition captures the most relevant aspect of bias, which is that it is "an inclination" "for or against one person or group". In its essence, bias is about differential treatment. Whether or not bias is unfair can only be defined by using a theory of justice. (Whether or not anything is unfair can only be defined using a theory of justice.) There is no single theory to rule them all, therefore in every use case, one must make a case for the theory of justice that one decides to employ. Therefore, instead of defining bias using a theory of justice at this point, we explain that this is the process. We explain this approach step by step in our paper. This approach itself is not based on a theory beyond the common methods of philosophical reasoning.
>
> How did you arrive at the figures in Tables 3 & 4? What is the exact methodology here?
> Each paper in this corpus has been assigned to categories, which include language, social bias and type of paper. In Table 3 (social biases) and 4 (languages) we computed the residuals between observed joints and predictions based on margins to show where there is clear oversaturation of work for specific combinations of languages (green) based on the current set of papers published at ACL venues.
>
> The analysis focuses on papers from the ACL anthology. Could you discuss any potential limitations or biases that might introduce, compared to a cross-venue survey?
> Yes, there are limitations when only looking at “one” publishing venue, such as (i) missing important work that was published at other venues (ii) not giving a holistic view of bias and fairness in NLP and (iii) categorizing and analyzing existing work that may have already focused on areas that are not represented in the ACL community. However, our goal in this review, is to characterize the research landscape within our community and highlight where there is opportunity to conduct more diverse research.
>
>
> Could you elaborate on how you envision your categorization and findings directly informing actionable next steps for the community?
> Our findings show in Figures 1 and 2 as well as Table 3 and 4 that there is a clear over- and underrepresentation of work on various social biases and languages, where the vast majority of work published at ACL focuses on English and gender bias. This is not inherently a bad thing, but it means that there is plenty of opportunity for growth in other areas. In the discussion section we give examples of biases and languages that are underrepresented and summarize how (as a community) we should aim to create a more inclusive and diverse research environment.

---

### Official Review · Reviewer_RQ6n · 2023-08-05

**Soundness:** 2

**Excitement:**

3: Ambivalent: It has merits (e.g., it reports state-of-the-art results, the idea is nice), but there are key weaknesses (e.g., it describes incremental work), and it can significantly benefit from another round of revision. However, I won't object to accepting it if my co-reviewers champion it.

**Paper Topic And Main Contributions:**

This paper introduces a survey of papers published in the ACL anthology that deal with social biases in different ways, for example by introducing datasets or methods to enable the detection of bias. The authors provide a categorisation of these paper with regard to what type of bias they are concerned with, what the contribution of the paper is and what languages the paper works on.

**Reasons To Accept:**

a) Interesting and relevant topic

b) Multiple dimensions considered (type of bias, type of paper, languages)

**Reasons To Reject:**

a) Narrow filtering of papers
The authors select papers from the ACL Anthology (which I think is ok, considering this is a survey of research in the NLP community which should be well-represented by the Anthology) filtered by a list of keywords. This keyword filtering seems somewhat too restrictive to me. The authors themselves give the example that titles like "fatphobia detection in online forums" would not be covered by their search terms. I think that the list could have been extended with words or word fragments like "-phob-", "-just-", "good" (rather than only "social good") and probably others without producing too many false positive, which are manually filtered in the next step anyways. Another possibility would be to train a classifier to detect whether a paper deals with bias, although this would of course introduce its own problems. However, it could be combined with the keyword approach to make sure that none of the currently covered papers are missed. Overall, I'm not sure whether the filtering process is suitable.

b) Findings could be structured better
The Empirical Findings section, providing the overview over existing research, is somewhat hard to read because it is missing some more structure. In the "Methods" and "Social biases" subsections, it would be extremely helpful to start with a description of the types of methods/biases that are analysed. Also, I feel like the paper mixes up the overview with the discussion of papers. The social bias subsetion also does not mention which types of biases are investigated, where these types come from or how the papers were assigned to a type

c) Discussion does not match reported results
The authors provide a good discussion of research on bias, which - in my opinion - raises valid and important points. However, it does not really build on the results reported in the previous section, but rather introduces new points that have not been analysed before:
- Suitable evaluation metrics are discussed, but have not been analysed in the previous section - although this would have been very interesting.
- The source of funding is stated as a potential problem, but not analysed before.
- The authors also state that only a small number of papers introduces new resources, while I think that the ratio of ~20% of all bias papers introducing resources reported previously is actually pretty good.
- The authors warn that gender bias frequently focuses on binary gender, but this has also not been analysed in the previous section.

Overall, I think that the premise of the survey is good, but there are some points that need to be addressed, as described above. Specifically, I think that the points raised in the discussion would have to be explicitly analysed in the previous section, which would also make the contribution of the overview section much stronger by providing more insight into the field.
My low score in reproducibility is due to the missing information on how papers were assigned to a bias/method type and partially the manual filtering process for the papers, which only mentions rough guidelines for which papers where discarded (the filtering is only a minor problem).



---
Update after Rebuttal: Since my weak points have not been addressed, I keep my scores unchanged.

**Reproducibility:**

2: Would be hard pressed to reproduce the results. The contribution depends on data that are simply not available outside the author's institution or consortium; not enough details are provided.

**Reviewer Confidence:**

3: Pretty sure, but there's a chance I missed something. Although I have a good feel for this area in general, I did not carefully check the paper's details, e.g., the math, experimental design, or novelty.

**Typos Grammar Style And Presentation Improvements:**

- l. 80: "outline[s]"
- l. 171: "papers[,] published"
- l. 196: "belong[s]"
- l. 197: "authors[']"
- l. 333: "matri[x]"
- l. 370: duplicate sentence
- Table 3: Wrong caption
- Figure 1: appears very late in the paper in relation to where it is mentioned
- Table 4: Second row/column has no caption. Also, "sp" = "es"?
- Section 5: While the number of bias papers does indeed go up in the table, the number of all papers in NLP has also increased a lot in these years. Could you add a percentage?

---

> ### Author Rebuttal · Authors · 2023-08-28
>
> Firstly, we would like to thank the reviewers for their kind and considered answers that will ultimately help us to improve our work. We have tried our best to answer the questions asked from all reviewers and cross-post them for all to see.
>
> What do you consider the main theoretical basis of your definition of bias?
> From the perspective of ethics and philosophy, bias definition has to be constructed in two steps: (1) what does "bias" mean and (2) what is an "unfair" bias. We take the dry definition of bias (which in itself does not necessarily imply an unfairness or negativity) from dictionary. The dictionary definition captures the most relevant aspect of bias, which is that it is "an inclination" "for or against one person or group". In its essence, bias is about differential treatment. Whether or not bias is unfair can only be defined by using a theory of justice. (Whether or not anything is unfair can only be defined using a theory of justice.) There is no single theory to rule them all, therefore in every use case, one must make a case for the theory of justice that one decides to employ. Therefore, instead of defining bias using a theory of justice at this point, we explain that this is the process. We explain this approach step by step in our paper. This approach itself is not based on a theory beyond the common methods of philosophical reasoning.
>
> How did you arrive at the figures in Tables 3 & 4? What is the exact methodology here?
> Each paper in this corpus has been assigned to categories, which include language, social bias and type of paper. In Table 3 (social biases) and 4 (languages) we computed the residuals between observed joints and predictions based on margins to show where there is clear oversaturation of work for specific combinations of languages (green) based on the current set of papers published at ACL venues.
>
> The analysis focuses on papers from the ACL anthology. Could you discuss any potential limitations or biases that might introduce, compared to a cross-venue survey?
> Yes, there are limitations when only looking at “one” publishing venue, such as (i) missing important work that was published at other venues (ii) not giving a holistic view of bias and fairness in NLP and (iii) categorizing and analyzing existing work that may have already focused on areas that are not represented in the ACL community. However, our goal in this review, is to characterize the research landscape within our community and highlight where there is opportunity to conduct more diverse research.
>
>
> Could you elaborate on how you envision your categorization and findings directly informing actionable next steps for the community?
> Our findings show in Figures 1 and 2 as well as Table 3 and 4 that there is a clear over- and underrepresentation of work on various social biases and languages, where the vast majority of work published at ACL focuses on English and gender bias. This is not inherently a bad thing, but it means that there is plenty of opportunity for growth in other areas. In the discussion section we give examples of biases and languages that are underrepresented and summarize how (as a community) we should aim to create a more inclusive and diverse research environment.

---

### Official Review · Reviewer_ysLf · 2023-08-11

**Soundness:** 2

**Excitement:**

2: Mediocre: This paper makes marginal contributions (vs non-contemporaneous work), so I would rather not see it in the conference.

**Paper Topic And Main Contributions:**

This paper presents a survey of 348 works on bias in the ACL Anthology. The authors find that work on bias is a growing field, with a majority of works focusing on bias detection, the English language, and gender bias. They additionally list a variety of issues in the field, such as the need for interdisciplinary research and the development of bias measuring metrics for downstream tasks, among others.

**Questions For The Authors:**

Question A: What do you consider the main theoretical basis of your definition of bias?
Question B: How did you arrive at the figures in Tables 3 & 4? What is the exact methodology here?

**Reasons To Accept:**

The paper shows broad knowledge of long-standing and emergent issues in the field of bias in NLP.
The authors moreover determined different biases while reviewing each paper, resulting in the contribution of a bias taxonomy of current bias research in NLP.  They also present a good overview of popular research topics in bias research. Showing the popularity of topics with numeric values like they did here can help shape the research direction of the community towards under-researched topics, and constitutes a valuable resource.

**Reasons To Reject:**

The main weakness of this paper is that it is missing structure, which makes it difficult to follow. This mainly concerns structural integrity within the individual sections (e.g. the _Findings_ section should be focused on findings of the **present**, not prior research), but also following common conventions for structuring papers (e.g., the paper is missing a _Conclusions_ section).
I believe that the missing division of prior work and present work negatively impacts the presentation of the contributions, which are hard to make out. Here, a tabular or graphical overview of the paper categories surveyed would have been helpful. The paper is further missing an overall, summarizing assessment of the direction of the field based on the present survey’s results, which is usually one of the main contributions of a survey.

**Reproducibility:**

2: Would be hard pressed to reproduce the results. The contribution depends on data that are simply not available outside the author's institution or consortium; not enough details are provided.

**Reviewer Confidence:**

4: Quite sure. I tried to check the important points carefully. It's unlikely, though conceivable, that I missed something that should affect my ratings.

**Typos Grammar Style And Presentation Improvements:**

- There were several instances of mixed up `\citep{}` where it should have been `\citet{}`. Please revise throughout. Examples: ll. 350, 436, 445

**Other Typos + Style issues**
- l. 016: principles
- l. 333: matrix
- ll. 366f.: look _at_ media bias _in an_ immigration dataset (MARDY) but _do_ not mention ...
- ll. 370f.: line repeated
- l. 397: Western and anglo-centric
- l. 446: _do not_ instead of contraction. Please revise throughout.
- Table 3 + 4 are figures, not tables

---

> ### Author Rebuttal · Authors · 2023-08-28
>
> Firstly, we would like to thank the reviewers for their kind and considered answers that will ultimately help us to improve our work. We have tried our best to answer the questions asked from all reviewers and cross-post them for all to see.
>
> What do you consider the main theoretical basis of your definition of bias?
> From the perspective of ethics and philosophy, bias definition has to be constructed in two steps: (1) what does "bias" mean and (2) what is an "unfair" bias. We take the dry definition of bias (which in itself does not necessarily imply an unfairness or negativity) from dictionary. The dictionary definition captures the most relevant aspect of bias, which is that it is "an inclination" "for or against one person or group". In its essence, bias is about differential treatment. Whether or not bias is unfair can only be defined by using a theory of justice. (Whether or not anything is unfair can only be defined using a theory of justice.) There is no single theory to rule them all, therefore in every use case, one must make a case for the theory of justice that one decides to employ. Therefore, instead of defining bias using a theory of justice at this point, we explain that this is the process. We explain this approach step by step in our paper. This approach itself is not based on a theory beyond the common methods of philosophical reasoning.
>
> How did you arrive at the figures in Tables 3 & 4? What is the exact methodology here?
> Each paper in this corpus has been assigned to categories, which include language, social bias and type of paper. In Table 3 (social biases) and 4 (languages) we computed the residuals between observed joints and predictions based on margins to show where there is clear oversaturation of work for specific combinations of languages (green) based on the current set of papers published at ACL venues.
>
> The analysis focuses on papers from the ACL anthology. Could you discuss any potential limitations or biases that might introduce, compared to a cross-venue survey?
> Yes, there are limitations when only looking at “one” publishing venue, such as (i) missing important work that was published at other venues (ii) not giving a holistic view of bias and fairness in NLP and (iii) categorizing and analyzing existing work that may have already focused on areas that are not represented in the ACL community. However, our goal in this review, is to characterize the research landscape within our community and highlight where there is opportunity to conduct more diverse research.
>
>
> Could you elaborate on how you envision your categorization and findings directly informing actionable next steps for the community?
> Our findings show in Figures 1 and 2 as well as Table 3 and 4 that there is a clear over- and underrepresentation of work on various social biases and languages, where the vast majority of work published at ACL focuses on English and gender bias. This is not inherently a bad thing, but it means that there is plenty of opportunity for growth in other areas. In the discussion section we give examples of biases and languages that are underrepresented and summarize how (as a community) we should aim to create a more inclusive and diverse research environment.

---

### Meta-Review · Area_Chair_xyy7 · 2023-09-16

**Recommendation:** 1

**Metareview:**

This manuscript provides an analysis and discussion on the state of fairness research made available through the ACL anthology. The authors find significant gaps as research predominantly is focused on gender bias and in English.

All three reviewers note the importance of this work, however two important concerns are raised which are not addressed by the authors in the author response:

1. The organisation of the manuscript makes it hard to read and hard to draw out findings from the survey
2. The keywords used to seed the paper selection may be too restrictive

While both warrant answers and additional work, the lack of clear presentatin suggests that the manuscript can use another round of editing.

---

### Decision · Program_Chairs · 2023-10-07

**Decision:**

Reject

**Comment:**

This manuscript provides an analysis and discussion on the state of fairness research made available through the ACL anthology. The authors find significant gaps as research predominantly is focused on gender bias and in English.

All three reviewers note the importance of this work, however two important concerns are raised which are not addressed by the authors in the author response:

1. The organisation of the manuscript makes it hard to read and hard to draw out findings from the survey
2. The keywords used to seed the paper selection may be too restrictive

While both warrant answers and additional work, the lack of clear presentatin suggests that the manuscript can use another round of editing.